# Synchrotron X-ray Scattering Analysis of Nylon-12 Crystallisation Variation Depending on 3D Printing Conditions

**DOI:** 10.3390/polym12051169

**Published:** 2020-05-20

**Authors:** Benjamin de Jager, Thomas Moxham, Cyril Besnard, Enrico Salvati, Jingwei Chen, Igor P. Dolbnya, Alexander M. Korsunsky

**Affiliations:** 1MBLEM, Department of Engineering Science, University of Oxford, Parks Road OX1 3PJ, UK; thomas.moxham@eng.ox.ac.uk (T.M.); cyril.besnard@eng.ox.ac.uk (C.B.); enrico.salvati@eng.ox.ac.uk (E.S.); jingwei.chen@eng.ox.ac.uk (J.C.); 2Dipartimento Politecnico di Ingegneria e Architettura, University of Udine, Via Delle Scienze, 33100 Udine, Italy; 3Diamond Light Source, Harwell Oxford Campus, Didcot OX11 0DE, UK; igor.dolbnya@diamond.ac.uk

**Keywords:** fused filament fabrication, polyamide 12 (PA12), crystallisation, differential scanning calorimeter, X-ray diffraction

## Abstract

Nylon-12 is an important structural polymer in wide use in the form of fibres and bulk structures. Fused filament fabrication (FFF) is an extrusion-based additive manufacturing (AM) method for rapid prototyping and final product manufacturing of thermoplastic polymer objects. The resultant microstructure of FFF-produced samples is strongly affected by the cooling rates and thermal gradients experienced across the part. The crystallisation behaviour during cooling and solidification influences the micro- and nano-structure, and deserves detailed investigation. A commercial Nylon-12 filament and FFF-produced Nylon-12 parts were studied by differential scanning calorimetry (DSC) and wide-angle X-ray scattering (WAXS) to examine the effect of cooling rates under non-isothermal crystallisation conditions on the microstructure and properties. Slower cooling rates caused more perfect crystallite formation, as well as alteration to the thermal properties.

## 1. Introduction

Fused filament fabrication (FFF) is an additive manufacturing (AM) method that produces objects by heating an extruded filament of material before putting it down in a line-wise and layer-wise fashion to build up the desired shape, allowing the adjacent tracks and layers to cool and start to bond before the deposition of subsequent lines and layers [1]. FFF allows for significant design flexibility and more efficient use of material compared to traditional methods and has seen increasing uptake in industries such as consumer goods, aerospace and biomedical technology [2]. Despite significant technological advances in recent years, the control over the outcome of the FFF process remains largely empirical, and the uncertain quality of the final product is still a barrier to wider industrial adoption. The underlying reason for this is a lack of full understanding of the link between the processing conditions, resulting microstructure, and the mechanical properties. A greater comprehension of the process-structure-property link is key to unlocking the technology’s application potential.

The process-structure-property link in FFF has several key components. On the microstructural level, the thermal history undergone by the filament on deposition affects the microstructure. In particular, the filament undergoes an initial cooling before being reheated by the deposition of subsequent layers. Bellehumeur et al. [3] and Srivinas et al. [4] found that the thermal history of an individual filament during FFF consists of a series of peaks whose amplitude decays with reheating, between which rapid cooling occurs. The adhesion between adjacent filaments, which may be called the mesostructure, is also affected by the thermal conditions during the bonding process, as reptation of polymer chains across the interface is required for layers to bond. Both the microstructural and mesostructural properties of an FFF sample will affect its mechanical performance.

The microstructure of a semi-crystalline polymer consists of its crystal structure. Crystallisation is a process which proceeds by both nucleation and growth, both of which have temperature dependence. The spatially varying crystal structure of FFF samples is therefore dependent on the specific thermal conditions of the build. The microstructural properties of Nylon-12 depend closely on the build temperature and the overall timescale [5], meaning that the specific build conditions have a significant effect on the spatially varying properties of the sample produced. Although FFF is a dynamical and complex process with many aspects to consider, the crystallisation behaviour of an individual filament can reveal details about the overall process.

Since it has been shown that an individual filament does not approach melting temperature *T*_m_ during subsequent reheatings, a significant part of the crystallisation during FFF occurs during the initial cooling process. Vaes et al. [6] showed through fast scanning chip calorimetry (FSC) of polyamide 6/66 that a significant proportion of obtained crystallinity in FFF samples is acquired during the initial heating and reheating cycles. As such, the bottom of each printed layer and the top of the one below each experience highly different non-isothermal processing conditions [7].

Nylon-12 is a semi-crystalline thermoplastic polymer with a high strength and stiffness that is one of the advanced polymeric materials used in FFF. It was selected for this study due to its specific thermal behaviour, which is characterized by a relatively large temperature interval between the start of melting, typically at 190 °C, and the start of crystallisation, between 160 and 170 °C [8]. This allows the material to be maintained at an elevated temperature after the melting of the filament during extrusion, which allows for minimizing the warpage. Depending on the processing conditions, Nylon-12 shows a complex polymorphism, whereby the γ-, γ’- and α-forms can arise [9]. The most stable form of Nylon-12 is the γ-form, which is obtained by cooling from the melt at atmospheric pressure, and is characterized by a monoclinic unit cell with the following parameters: *a* = 9.33 Å, *b* = 16.88 Å, *c* = 4.78 Å and β = 120° [10]. The γ’-form consists of smaller crystals of the same structure as the γ-form with many defects and transforms into the γ-form on annealing [11].

This study focuses on a particular aspect of the processing-structure-property relationship in FFF samples, namely the relationship between thermal history in the form of cooling rates and crystal structure. The rapid cooling rates experienced by individual filaments during the printing process are considered to be a non-isothermal process. Non-isothermal crystallisation was performed on Nylon-12 filament samples both in situ with wide-angle X-ray scattering (WAXS) and ex situ through differential scanning calorimetry (DSC). The data from these tests were compared to the analysis of FFF parts produced under different thermal conditions. In addition, the effect of the cooling rate from print chamber temperature to room temperature on the crystal structure of FFF samples was examined.

## 2. Materials and Methods

The experimental results reported in this paper were collected during two experimental sessions in which X-ray scattering and DSC were used to characterise Nylon-12 samples. In the first session, 3D-printed samples produced by FFF were characterised. In the second session, Nylon-12 filament was heated to the melting temperature, and then non-isothermally crystallised from melt.

In order to provide clarity with regard to the provenance of the samples, these are listed at the beginning of each results section.

Nylon-12 filament manufactured by Breathe3DP was used to produce samples using an RS Pro IdeaWerk 3D printer (RS Components, Corby, UK) and a P220 printer (Apium GmbH, Karlsruhe, Germany). Both printers used an extruder temperature of 230 °C. RS Pro samples (henceforth “FFF RS”) were manufactured with the print bed and environment held at room temperature. Apium samples (henceforth “FFF Apium”) were manufactured with the print bed and environment held at 120 °C before being slowly cooled at 2 °C/min to room temperature. In addition, samples produced by the Apium P220 were either allowed to cool slowly (henceforth “slow-cooled”) or removed from the printer and cooled rapidly (henceforth “air-cooled”).

All FFF-manufactured samples were built according to the same printing path, with parts having a 100% internal fill. The internal fill path was a diagonal cross-hatch. The deposition speed, which in FFF can influence the temperature and viscosity of the deposited filament [12], was kept constant for both samples. This minimises the effect of deposition-induced orientation along the filament direction on the crystal structure.

DSC analysis was carried out under nitrogen atmosphere on Nylon-12 samples using a TA Instruments DSC Q2000 (TA Instruments, New Castle, DE, USA) to evaluate the thermal properties and assess the crystallisation behaviour of the samples. The sample weight was between 2.5 and 3.5 mg. Heating and cooling cycles were performed on FFF-produced Nylon-12 samples between 20 and 225 °C at a ramp rate of 10 °C/min. To simulate the cooling conditions during FFF, non-isothermal crystallisation was performed as follows: Nylon-12 filament was heated at 10 °C/min to 230 °C and held for 5 min to eliminate residual crystals. Then the melt was cooled at different cooling rates: 2, 5, 10, 20 and 40 °C/min, respectively, to 40 °C, before being reheated to melt. The thermal profile of the tests is shown in Figure 1.

In situ wide angle X-ray scattering (WAXS) was carried out at the Diamond Light Source synchrotron (B16 beamline, Harwell Oxford Campus, Didcot, UK) to identify the crystalline phases and observe their response to thermal conditions. An ImageStar 2D diffraction detector (Photonic Science Ltd., Saint Leonards, UK) was used to register the scattering patterns from the X-ray beam with wavelengths of 0.8277 Å (15keV) for the first session and 0.6898 Å (18keV) for the second session. A Linkam TS1500 furnace (Linkam Scientific, Epsom, UK) was used to heat the Nylon-12 filament from room temperature to 230 °C at a ramp rate of 10 °C/min. A custom aluminium sample holder fastened by Capton was used to mount the samples in the furnace. Following this, the filament samples were cooled at the prescribed rates of 5, 10, 20 and 40 °C/min analogous to DSC testing. Nylon-12 samples produced by FFF under differing thermal conditions were also characterised in situ using WAXS.

The WAXS data were analysed using DAWN Science (Diamond Light Source, Harwell Campus, Didcot, UK). The pixel to degree conversion was obtained through calibration with LaB6 sample. For each principal crystalline reflection in the data, the integrated intensity and peak position were obtained by fitting a pseudo-Voigt function. For the amorphous component, a sum of additional broad Gaussian functions was needed. An estimate of the crystallinity *C* was obtained by dividing the sum of the intensities of the crystalline reflections by the total intensity as per Equation (1):(1)$$C=\frac{\sum I_{c}}{\sum I_{c}+\sum I_{a}}$$
where *I*_c_ and *I*_a_ are the crystalline and amorphous integrated intensities, respectively.

## 3. Results

### 3.1. Differential Scanning Calorimetry (DSC) of Nylon-12

Table 1 shows a full list of samples tested in this section, along with their sources or method of production.

The results for the non-isothermal measurements are shown in Figure 2. It can be observed that a slower cooling rate leads to a shifting of the crystallisation onset temperature *T*_ic_ towards a higher temperature. The peak of the crystallisation exotherm is also shifted considerably towards higher temperatures, while the magnitude of the exotherm peak decreases. The results for the calculated crystallisation halftime *t*_1/2_ are included in Table 2.

Figure 2a shows the melting endotherms during sample reheating after non-isothermal crystallisation. It can be observed that there is a continual evolution of the shape depending on the attendant cooling rate. At 2 °C/min, the endotherm shows a single peak, whereas a double peak appears at faster cooling rates. In particular, the ratio of the magnitudes of the peaks within the doublet alters as a function of the cooling rate. This is plotted as a function of the cooling rate in Figure 3. The primary melting endotherm, as shown most clearly in the curve obtained at 2 °C/min, is designated as Peak I. This can be seen to shift to a progressively lower temperature as the cooling rate increases to 40 °C/min. The secondary melting endotherm, which manifests at cooling rates above 10 °C/min, is designated Peak II. Figure 2c,d show the percentage of crystallinity *X*(*t*) values obtained as functions of temperature and time. The curves become flatter in the later stage of crystallisation. This can be attributed to spherulite impingement and is discussed in more detail later. Figure 2c shows that even for the fastest cooled sample, the bulk of the crystallisation is complete by 150 °C, with most of the crystallisation taking place above 170 °C for the slower cooled samples. Table 2 lists the peak temperature *T**, the crystallisation halftime *t*_1/2_ and the crystallisation enthalpy Δ*H*_c_ for the different cooling rates.

Figure 4 demonstrates the thermal behaviour of the FFF filament and 3D-printed parts. In all three samples, a large endotherm is present at the temperature of 197 °C, the magnitude of which is larger for the FFF samples than for the filament. In addition, a shoulder is seen on the low-temperature side for the FFF-produced samples, which is not present for the filament.

### 3.2. In Situ Wide-Angle X-ray Scattering (WAXS)

Table 3 shows a full list of test samples described in this section, along with their sources or method of production.

Figure 5 shows the 2D WAXS patterns collected at 25 °C following the completion of crystallisation for all cooling rates. Although samples were set to cool at rates equal to the cooling rates during the non-isothermal DSC experiments, the necessity of pausing the cooling temperature ramp in order to acquire WAXS patterns resulted in the true cooling rates differing from the nominally prescribed cooling rates. The nominal and actual cooling rates are presented in Table 4. For clarity of attribution, in the discussion below these samples are referred to as Samples I-IV in the order of the true cooling rates that are seen to span more than one decade of magnitude. The data for the percentage of crystallinity and the crystal lattice *d*-spacing calculated from the WAXS patterns are presented in Figure 6.

Figure 5 reveals that two diffraction peaks are present for all cooling rates: the principal peak at the scattering angle 2*θ* = 9.2° (corresponding to the interplanar lattice spacing of 4.34 Å), attributable to the 001(+20$\bar{1}$) reflection [10] and designated as Peak I, along with a lower intensity peak at a higher 2*θ* (closer interplanar spacing), attributed to the 200 reflection and designated as Peak II. Both peaks are surrounded by a broad halo that arises due to scattering from the amorphous regions within the sample. These peaks are annotated in Figure 5a.

Significant differences can be observed in the shape and position of the second peak depending on the cooling rate. In Sample I, Peak II is manifested as a shoulder found to be positioned at 2*θ* = 10.16° (3.89 Å). In Sample II, there is a more pronounced Peak II centred at the position 2*θ* = 10.48° (3.77 Å). In Sample III, Peak II appears again as a shoulder with the peak centre at 10.28° (3.84 Å). Finally, in Sample IV, Peak II is more pronounced still, and is positioned at 2*θ* = 10.62° (3.72 Å).

Figure 6 shows the behaviour of the two peaks during cooling for all four samples. It can be observed that in Samples I-III, there is an initial decrease in the *d*-spacing of Peak I due to thermal contraction prior to the emergence of Peak II. Subsequently, the *d*-spacing of Peak II continues to decrease in line with thermal contraction, while the *d*-spacing of Peak I increases. In Sample IV, the *d*-spacing of Peak I appears to increase throughout cooling. It is hypothesised that this is due to the slower cooling rates and greater time spent at the “normal” crystallisation temperature range of Nylon-12 (160–170 °C), allowing more perfect crystals to form from the onset. In addition, it can be observed that as the cooling rate slows, the *d*-spacing of Peak II decreases.

Figure 7 shows the evolution of the 1D WAXS patterns during cooling as a function of temperature, extracted from 2D detector data using radial binning. Following cooling below the melting temperature *T*_m_ = 197 °C, a single sharp peak is seen at 2*θ* = 9.52° (4.15 Å), which then gradually decomposes into the 001(+20$\bar{1}$) peak at 4.34 Å and the 200 peak at 3.72 Å. With decreasing temperature, the crystallinity increases continuously.

Figure 8 shows the 2D and 1D WAXS data from (a) the filament, (b) FFF RS and (c) FFF Apium samples. It can be observed that in the filament scattering pattern only one reflection is present corresponding to the 001(+20$\bar{1}$) reflection. In the FFF samples, the double peak pattern is seen, showing that the 200 reflection only appears following melting and recrystallisation. In particular, the slower-cooled FFF Apium sample shows a considerably more defined 200 peak with a smaller *d*-spacing than the FFF RS sample. Figure 9 shows the WAXS patterns for the slow-cooled and air-cooled samples (III and IV), where minimal differences in the patterns can be observed.

In the final experiment reported here, FFF Apium samples were heated from room temperature to 160 °C at a ramp rate of 5 °C/min and cooled back to room temperature at the same rate for two successive cycles. Significant changes in the sample microstructure reflected in the XRD patterns were observed. Figure 10 shows the evolution of the 2D WAXS patterns of the data as a function of increasing temperature.

## 4. Result Analysis

### 4.1. On the Crystallisation Structures of Nylon-12 Filament and FFF-Produced Samples

The analysis of the WAXS data with the accompanying calculation of *d*-spacing illustrated in Figure 8 shows that the Nylon-12 is in the γ-form throughout. However, the peak positions shown indicate that the microstructure is distinct from the standard γ-form Nylon-12 for which a single reflection at 4.15 Å would be expected, corresponding to the 200 plane, here a single reflection at 4.34 Å corresponding to the 001(+20$\bar{1}$) reflection is observed, along with the second peak corresponding to the 200 position after melting.

The differences seen in the WAXS patterns in Figure 5 demonstrate that the FFF process has a significant effect on the structure of Nylon-12. In particular, the emergence of a second diffraction peak after melting indicates a rearrangement of the crystalline structure. The emergence of an additional diffraction peak due to heating has been observed in Nylon-6 as a result of the Brill transition [13], whereby the pseudohexagonal structure transforms to a monoclinic one. However, previous research on Nylon-12 and other even-even nylons has shown no evidence that a Brill transition takes place [14]. It is therefore proposed that this alteration in the Nylon-12 WAXS pattern post-melt is the result of bulk isotropic alignment and realignment of the unit cells.

The unit cell of Nylon-12 depicted in Figure 11 shows that it is formed from parallel polymer chains with H-bonds between the N and O atoms. These H-bonded chains form a sheet, with the arrangement of these sheets in a 3D structure forms the complete unit cell. The H-bonds between chains exist along the *c*-axis. In Figure 11a, a single diffraction ring with the *d*-spacing of 4.34 Å corresponding to the 001(+20$\bar{1}$) reflection is observed that corresponds to the 4.78 Å spacing of the unit cell *c*-axis. This indicates that there is bulk isotropic alignment with a selective uniplanar orientation along the *c*-axis. This alignment is due to the extension of the long-chain Nylon-12 molecules in the flow field associated with filament production, whereby the chains (corresponding to the *b*-axis) orient themselves parallel to the filament direction, acting as row nuclei for subsequent lamellar growth [15]. In addition, the Nylon-12 undergoes shear force in the nozzle during the deposition process and subsequent stretching during deposition, causing additional regular chain structure and the formation of microfibres [16].

Based on our observations, a possible picture of the internal structure evolution is proposed here. Following melting and recrystallisation, the chain alignment within the filaments is preserved; however, the lamellae are reorganized into a more energetically favourable state such that the lamellar sheets are realigned. This causes additional planar orientation along the *a*-axis of the unit cell, causing the appearance of the 200 reflection at 3.72 Å. This alignment and realignment of the unit cell is illustrated in Figure 11.

It is notable that the positions of both peaks change depending on the cooling rates. The inter-sheet 200 reflection position at 3.72 Å is governed by the lamellar properties. DSC data show# that the less-perfect lamellae are formed at faster cooling rates, leading to a higher inter-sheet spacing. The alteration in the position of the principal 001(+20$\bar{1}$) reflection is likely to be driven by a distortion of the unit cell, corresponding to a change in the angle between amide planes.

Figure 4 and Figure 10 elucidate the nature of peak splitting. Figure 10 shows the WAXS patterns during in situ heating of the FFF Apium sample, where it can be observed that the 200 peak coalesces with the 001(+20$\bar{1}$) peak as the temperature increases above glass transition temperature *T*_g_ = 97 °C, which indicates a gradual increase in the *d*-spacing before the merger of the two peaks. Likewise, Figure 4 demonstrates that *a*-axis spacing that governs the 200 reflection position alters depending on cooling rate. It can therefore be inferred that the 200 reflection is produced by smaller, less perfect crystallites which melt at a lower temperature, since the 200 reflection is produced by the periodicity of the hydrogen bonding surface in the unit cell [17]. Rising temperature causes the lamellar stack period to increase, increasing the *d*-spacing; ultimately the periodicity in the *a*-axis is disrupted by the onset of melting of the less perfect crystallites. Further corroboration of this conceptual picture will be required.

### 4.2. Non-Isothermal Crystallisation as a Basis for FFF Processs Calibration

The analysis of the WAXS data presented above indicates that non-isothermal process analysis provides a good for understanding the material structure evolution during FFF. This is verified by the fact that the patterns seen in Figure 5 for Sample I display the same characteristics as the fast-cooled FFF RS sample, whereas the patterns for Sample IV are characteristically equivalent to the FFF Apium sample. This is due to the imperfect formation of crystallites during the fast cooling process. There is evidence to support the notion that both Sample I and the FFF RS sample have crystallized into the γ’-form, which consists of small crystals with many defects [18].

However, the DSC results also demonstrate the limitations of non-isothermal crystallization as a basis for FFF process calibration. The results from the DSC non-isothermal crystallization show that a faster cooling rate results in a lower initial melting temperature due to poorly formed crystallites. The single endotherm observed for the slower cooled samples indicates the presence of a well-formed primary crystal structure, whereas the double endotherm corresponds to the poorly formed crystals observed in Sample I. The lower-temperature endotherm indicates a lamellar reordering process, whereby thicker crystals are formed by the partial melting and recrystallisation of the thin lamellae. The melting of the primary crystal structure is indicated in Figure 2 as Peak I, whilst Peak II corresponds to the melting of secondary lamellar structures formed by melting and recrystallisation.

It is interesting to note that the FFF RS sample and Sample I produce WAXS patterns that are both qualitatively and quantitatively close. They may therefore be expected to give similar DSC traces with a double melting endotherm. However, unlike Sample I, the FFF RS sample displays a single melting endotherm with a small shoulder. Furthermore, the FFF RS sample is found to melt at 197 °C, whereas Sample I displays the principal endotherm at 195.03 °C. In addition to this, the FFF Apium sample which displays a WAXS pattern corresponding to Sample IV would be expected to display a single melting endotherm at around 192 °C. Instead, it shows a melting endotherm at 197.90 °C with a small lower-temperature endotherm that is quantitatively similar to that of the FFF RS. These observations highlight the fact that further differences may be present between samples in the nanoscale internal arrangement of the crystalline and amorphous phases which may induce significant disparity in the macroscopic thermal and mechanical properties.

### 4.3. Conclusions Regarding Nylon-12 FFF Processing Parameters

From these data, it can be concluded that the cooling rate is a key processing parameter that should be varied in order to obtain a better quality of crystallisation in Nylon-12 parts. The cooling rate in FFF is something that can be controlled by a number of measures. Primarily, these are the extrusion rate and the temperature difference Δ*T*_ext_ between the extrusion temperature and the ambient chamber temperature. It has previously mentioned that the extrusion rate can result in deposition-induced effects regarding the alignment of the polymer chains. Since this study focuses on thermal history versus crystal structure, Δ*T*_ext_ is the parameter recommended for variation.

The principal crystal structure formed during the FFF of Nylon-12 is the lamellar structure responsible for the 200 reflection. This structure can be contrasted to that responsible for the 001(+20$\bar{1}$) reflection, which is due to the chain alignment in the filament direction. It can therefore be deduced that improving the quality of this reflection corresponds to more perfect crystals consolidated in the print. Minimising Δ*T*_ext_, therefore, allows for a slower cooling rate of deposited filaments in the early stages of the print, allowing for more perfect crystals to form. In addition, it also raises the minimum temperature experienced by deposited filaments during the print, therefore allowing for a more gradual slower cooling rate during the process, and therefore a higher quality of crystallisation.

## 5. Discussion

Based on the analysis of the results presented in the previous section, several conclusions can be drawn. Principally, the limitations of non-isothermal crystallisation as a basis for FFF are shown, as the DSC data show that the thermal properties of the FFF-produced samples are distinct from those produced by non-isothermal crystallisation. This divergence in properties is proposed to be due to the range in printer chamber and bed temperatures attained during sample production.

The DSC data shown in Figure 2c demonstrate that the majority of the crystallisation is completed by the time the samples have cooled to 150 °C. Since the chamber temperature for the FFF Apium samples was set at 120 °C, it can be considered that these samples spend a majority of time during the print process at a temperature close to 120 °C. Individual filaments in FFF-produced samples will tend towards the chamber/bed temperature, with reheatings caused by adjacent layer deposition prolonging this cooling process. However, in both of the types of FFF sample analysed, the temperature range for this cooling process can be considered to be below the range of crystallisation temperatures for Nylon-12.

Therefore, the majority of the crystallisation undergone by the FFF samples is proposed to be the crystallisation during the initial deposition of the filament. This accounts for variation in the WAXS patterns of the FFF Apium and FFF RS samples, as the difference in chamber temperatures (120 °C vs. 25 °C respectively) leads to a significant difference in cooling rate following initial deposition. This can be considered on its own as being a non-isothermal crystallisation. However, as evidenced by the difference in the shape of the melting endotherms in the DSC data for the non-isothermally crystallised samples and the FFF-produced samples, it is not sufficient to describe the crystallisation processes during FFF.

It is hypothesized that this difference is due to two factors: firstly, the deposition-inducing effect, and secondly the effect of the cooling rates in the mesostructure. During the printing process, the filament undergoes additional shear and therefore chain alignment. While this is not expected to affect the unit cell of the γ-form, it may alter the crystallisation behaviour because regular chains are more likely to crystallise. Secondly, the reheatings due to subsequent layer deposition are hypothesised to raise the filament temperature sufficiently in the initial stages to allow for more complete crystallisation to occur. This would affect the thermal properties without being sufficient to transform the γ’-form in the FFF RS samples to the γ-form.

## 6. Conclusions

The suitability of the non-isothermal model as a basis for FFF process calibration can be summarised as follows:The variation of the WAXS patterns as a function of the cooling rate shows that it has a significant effect on the crystal microstructure of the Nylon-12 samples.Figure 2c demonstrates that the majority of crystallisation during the cooling process takes place by the time the sample has cooled to 150 °C, which is considerably higher than any chamber or bed temperatures attained during this experiment.The cooling rate at the later stages of the build has little to no effect on the microstructure, as demonstrated by the similarities in DSC patterns between the FFF Apium and FFF RS samples, and in the WAXS patterns between the slow-cooled and air-cooled samples.Further refinement of the model must be driven by the evaluation and control of the cooling rate during the print itself that is governed by the chamber temperature and the rate of extrusion.An increase in the chamber temperature closer to the crystallisation range for Nylon-12 (150–170 °C) will necessarily involve a shift towards an isothermal crystallization regime at constant chamber temperature, with temperature fluctuations due to the deposition of additional layers that can be thought of as a quiescent sinusoidal wave. This repeated cooling after reheating may nevertheless be considered as a non-isothermal process.

Further research is anticipated to focus on further control of the cooling rate via the selection of the chamber temperature, and the consideration of its effects on the microstructural and mechanical properties. This will assist with the development of a quantitative model to allow more accurate description of the crystallization process during the FFF of Nylon-12.

## Figures and Tables

**Figure 1 polymers-12-01169-f001:**
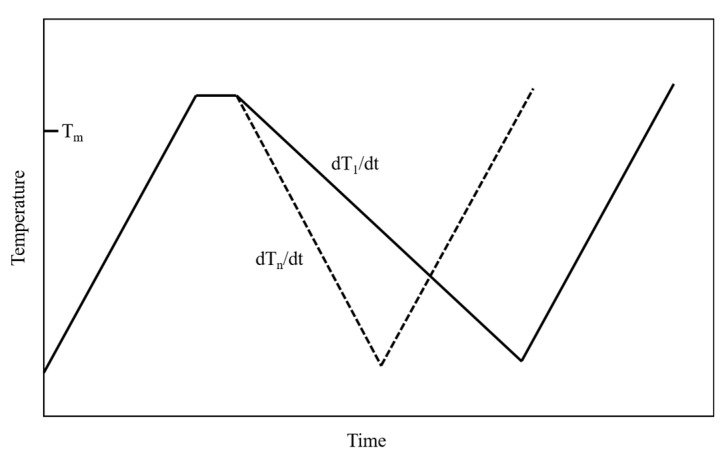
Schematic diagram of the temperature profiles used in the non-isothermal crystallisation DSC analysis. dT_i_/dt represents the cooling rate during the experiments.

**Figure 2 polymers-12-01169-f002:**
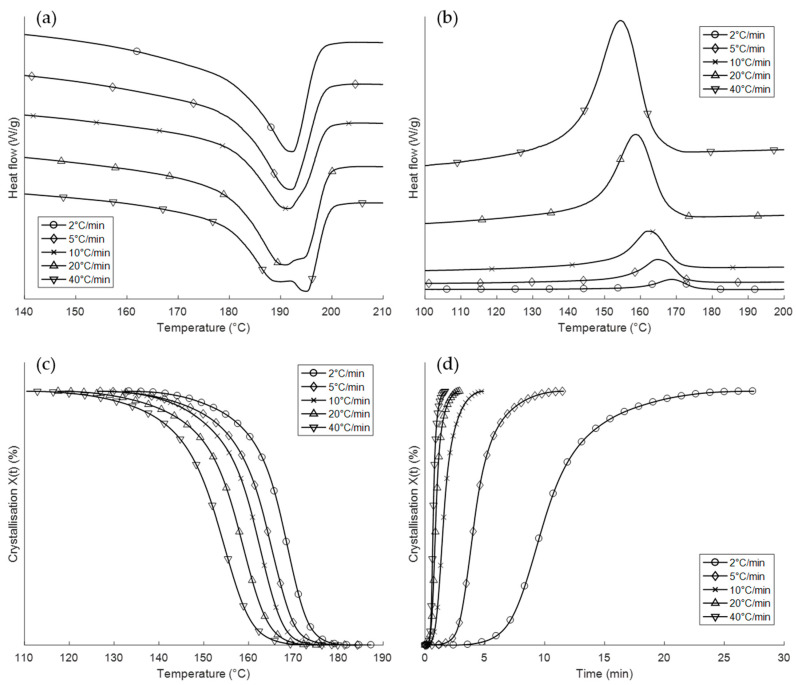
Results of the non-isothermal crystallisation experiments. (**a**) Heating endotherms for the second heating cycle. (**b**) Cooling exotherms during crystallisation. (**c**) Percentage of crystallinity *X*(*t*) as a function of temperature. (**d**) Percentage of crystallinity *X*(*t*) as a function of time.

**Figure 3 polymers-12-01169-f003:**
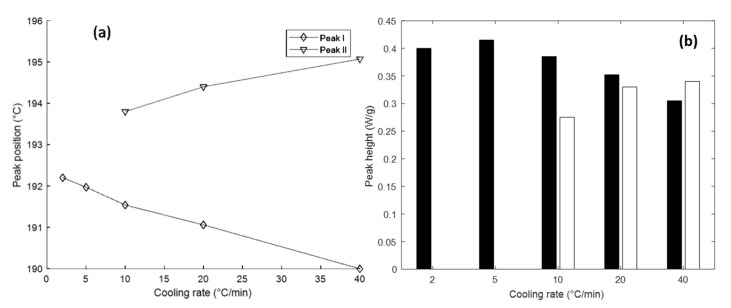
Peak positions and magnitude ratios for the non-isothermal crystallisation: (**a**) the evolution of peak centre temperature values, and (**b**) peak heights as a function of the cooling rate.

**Figure 4 polymers-12-01169-f004:**
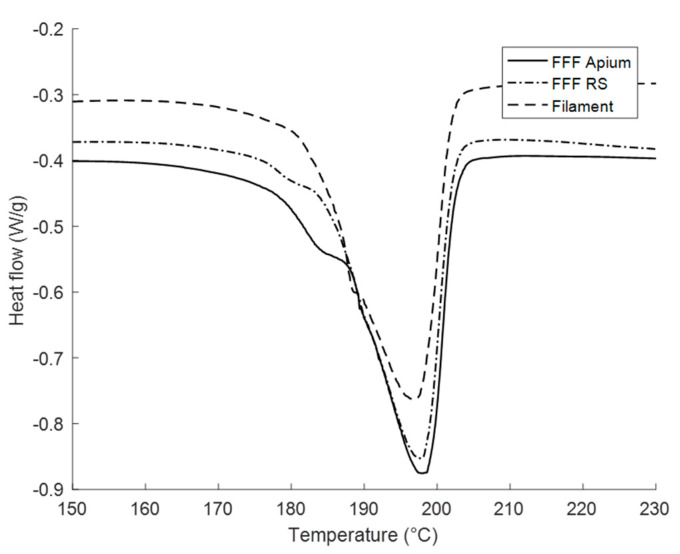
DSC thermograms for the first heating cycle of Nylon-12 filament, fused filament fabrication (FFF) Apium and FFF RS samples.

**Figure 5 polymers-12-01169-f005:**
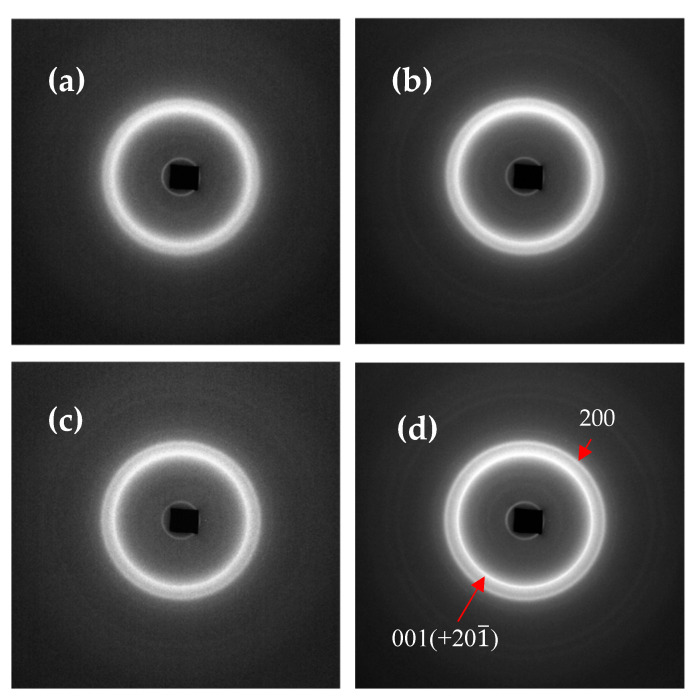
Two-dimensional WAXS patterns for the in situ non-isothermal crystallization experiments. (**a**) Sample I, (**b**) Sample II, (**c**) Sample III and (**d**) Sample IV. Inner and outer peaks were found to be due to background diffraction from the sample holder.

**Figure 6 polymers-12-01169-f006:**
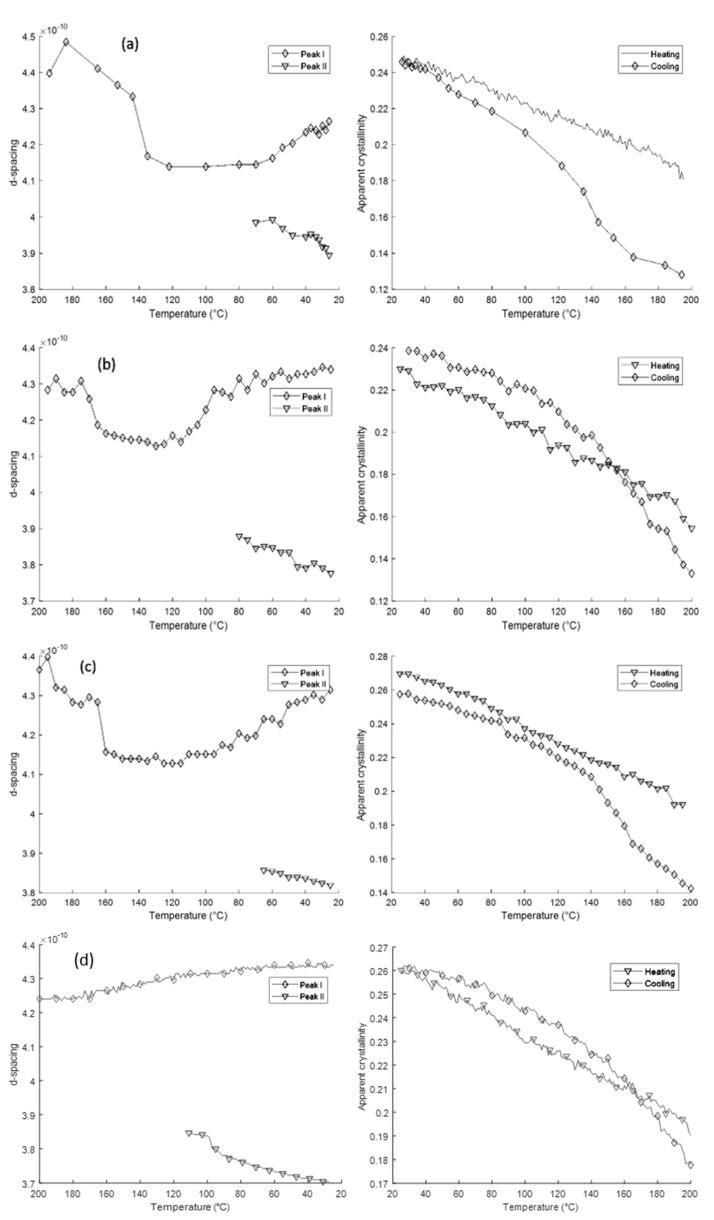
The evolution of the lattice *d*-spacing during cooling for the two peaks present shown alongside the percentage of crystallinity derived from X-ray analysis during heating and cooling. (**a**) Sample I, (**b**) Sample II, (**c**) Sample III, (**d**) Sample IV.

**Figure 7 polymers-12-01169-f007:**
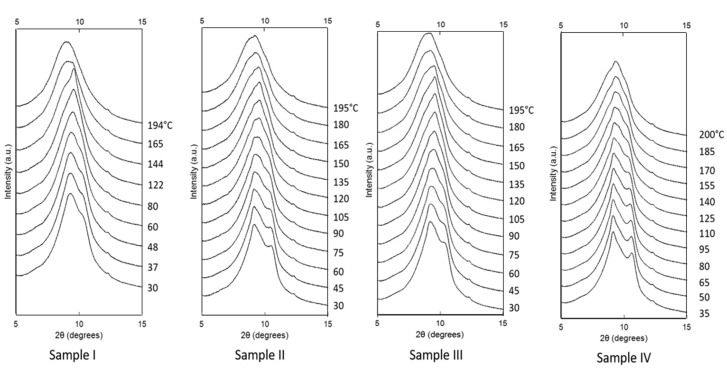
One-dimensional WAXS patterns (shifted vertically for clarity) for the four samples as a function of temperature during cooling. The small peak at 2θ = 12.5 °C is due to background diffraction.

**Figure 8 polymers-12-01169-f008:**
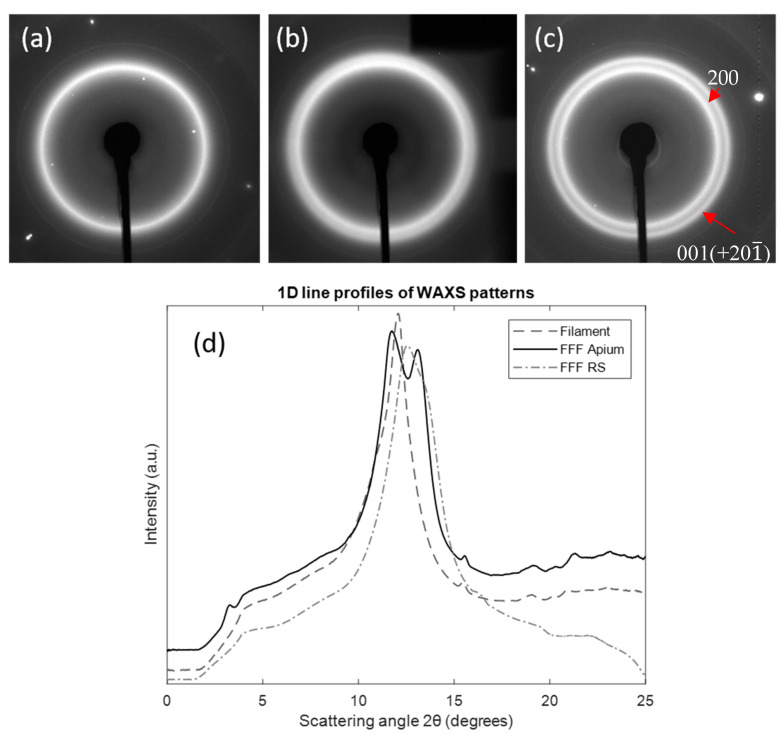
Two-dimensional and 1D WAXS patterns of the Nylon-12 samples. (**a**) Filament pattern, with only the 001(+20$\bar{1}$) peak visible. (**b**) FFF RS sample, with the shoulder corresponding to the 200 peak. (**c**) FFF Apium sample with two peaks clearly visible. (**d**) shows the 1D line profiles plotted as a function of diffraction angle 2θ. The dark shadow in (**b**) is due to partial beam blockage from a modified sample holder.

**Figure 9 polymers-12-01169-f009:**
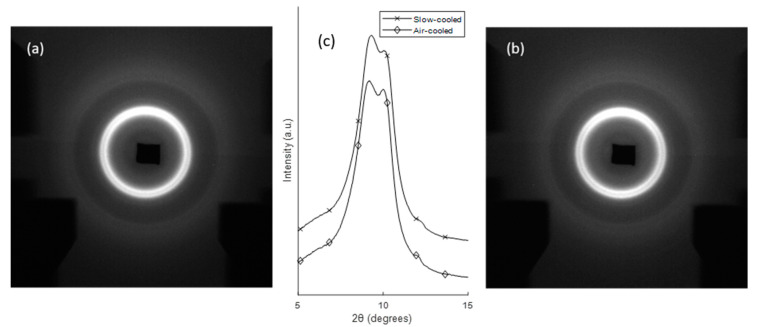
WAXS patterns of (**a**) FFF Apium slow-cooled sample (III), (**b**) FFF Apium air-cooled sample (IV). (**c**) Vertically shifted and overlaid 1D WAXS patterns extracted for both samples for comparison. The dark shadows in (**a**,**b**) are due to partial beam blockage from the sample holder.

**Figure 10 polymers-12-01169-f010:**
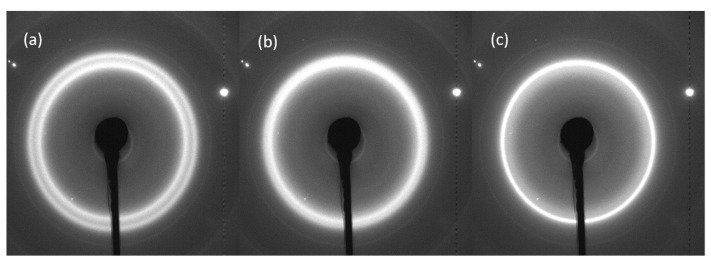
The evolution of the principal peaks in FFF Apium Nylon-12 samples. Note the coalescence of the double peaks into a single peak, presumed to be 001(+20$\bar{1}$), as *T* is increased. (**a**) The pattern at *T* = 25 °C. (**b**) The pattern at *T* = 65 °C showing the ongoing process of peak coalescence. (**c**) The pattern at *T* = 124 °C showing full peak coalescence. The bright spots and inner peak were found to be due to background diffraction from the sample holder.

**Figure 11 polymers-12-01169-f011:**
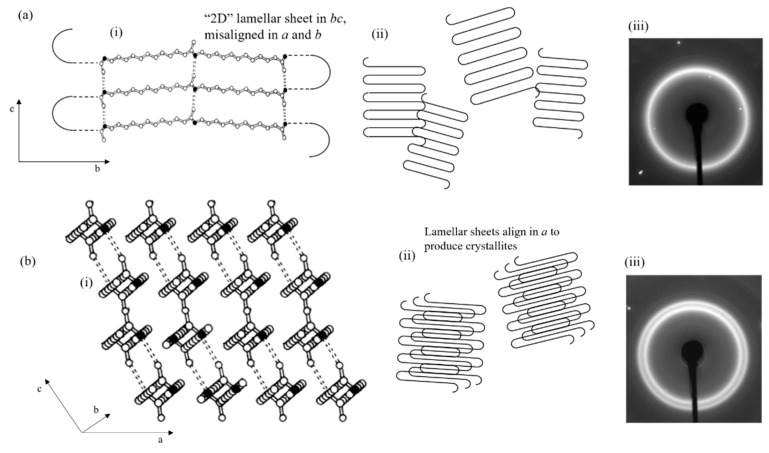
(**a**) Lamellar sheet in Nylon-12: (i) aligned chains bonded in the *c*-axis by H-bonds, (ii) conceptual diagram of misoriented lamellae, (iii) the resulting diffraction pattern with a single ring observed due to bulk isotropic alignment in *c*. (**b**) Lamellar sheets form energetically favourable planes along the *a*-axis, leading to a more complete unit cell and causing the emergence of the 200 diffraction ring. (i) Unit cell in full lamellar form, (ii) crystallites formed as a result, and (iii) the resulting diffraction pattern. Basic unit cell schematic of Nylon-12 according to Inoue and Hoshino [8].

**Table 1 polymers-12-01169-t001:** List of samples tested by DSC.

Sample Designation	Origin	Notes
40 °C/min	Nylon-12 filament	Non-isothermally crystallised at 40 °C/min
20 °C/min	Nylon-12 filament	Non-isothermally crystallised at 20 °C/min
10 °C/min	Nylon-12 filament	Non-isothermally crystallised at 10 °C/min
5 °C/min	Nylon-12 filament	Non-isothermally crystallised at 5 °C/min
2 °C/min	Nylon-12 filament	Non-isothermally crystallised at 2 °C/min
Filament	Drawing	Slow cooling during drawing
FFF Apium	FFF (Apium P220), *T*_chamber_ = 120 °C	Cooled slowly to room temp post print
FFF RS	FFF (RS Pro Ideawerk), *T*_chamber_ = 25 °C	Fast cooling rates throughout

**Table 2 polymers-12-01169-t002:** The kinetic parameters for the non-isothermal crystallisation of Nylon-12.

Ф (Cooling Rate)	2 °C/min	5 °C/min	10 °C/min	20 °C/min	40 °C/min
*T** (°C)	168.71	164.83	162.16	158.82	154.48
*t*_1/2_ (min)	10.04	4.17	1.56	0.91	0.69
Δ*H*_c_ (J/g)	36.51	36.77	35.49	33.55	30.08

**Table 3 polymers-12-01169-t003:** List of samples tested by wide-angle x-ray scattering (WAXS).

Sample Designation	Origin	Notes
Sample I	Nylon-12 filament	Non-isothermally crystallised (see Table 4)
Sample II	Nylon-12 filament	Non-isothermally crystallised (see Table 4)
Sample III	Nylon-12 filament	Non-isothermally crystallised (see Table 4)
Sample IV	Nylon-12 filament	Non-isothermally crystallised (see Table 4)
Filament	Drawing	Slow cooling during drawing
FFF Apium	FFF (Apium P220), *T*_chamber_ = 120 °C	Cooled slowly to room temp post print
FFF RS	FFF (RS Pro), *T*_chamber_ = 25 °C	Fast cooling rates throughout
Slow-cooled	FFF (Apium P220), *T*_chamber_ = 120 °C	Cooled slowly to room temp post print
Air-cooled	FFF (Apium P220), *T*_chamber_ = 120 °C	Cooled rapidly to room temp post print

**Table 4 polymers-12-01169-t004:** Nominal and actual cooling rates for Nylon-12 samples tested non-isothermally in situ.

Sample Designation	Nominal Cooling Rate (°C/min)	Actual Cooling Rate (°C/min)
Sample I	40	33.16
Sample II	20	12.38
Sample III	10	6.29
Sample IV	5	1.47

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
