# Peer review of "Synchrotron X-ray Scattering Analysis of Nylon-12 Crystallisation Variation Depending on 3D Printing Conditions"

_polymers, 2020, doi:10.3390/polym12051169_

Round 1

Reviewer 1 Report

The paper gives a very interesting insight into the structural behaviour of Nylon-12 during the printing via FFF. Although a final recommendation for users of such materials is lacking at the moment, it gives important information on what to consider when designing similar evaluation routines and when being a frequent user of Nylon-12.

The reviewer main concern is that the authors have not yet fully found the reason for the structural changes of Nylon-12 that are occurring during the printing process in comparison to just thermocycling the material. Although, they state that this part of ongoing or upcoming work, the reviewer recommend the authors to hypothesize speculation trying to explain the underlying phenomenon based on the available information.

There are 1 or 2 minor spelling errors throughout the paper and the thermogramm (Figure 4) is standing out since it seems to be plot from the measurement device itself rather than a postprocessed figure. If possible, please match to the style of the other diagrams.

In addition, they could improve on the formatting a bit, since the reader has to scroll through the document quite a lot to find the diagrams that are described in the text. This is something to be considered from a formatting point of view.

Author Response

We would like to sincerely thank all reviewers for taking the time to read our manuscript and provide constructive feedback. Based on these comments we have made a number of adjustments to improve the manuscript. All of these changes have been marked in yellow in the manuscript.

Reviewer 1

  • The paper gives a very interesting insight into the structural behaviour of Nylon-12 during the printing via FFF. Although a final recommendation for users of such materials is lacking at the moment, it gives important information on what to consider when designing similar evaluation routines and when being a frequent user of Nylon-12.

Thank you. We appreciate the need to seek final design recommendations, and continue our efforts in this regard.

  • The reviewer main concern is that the authors have not yet fully found the reason for the structural changes of Nylon-12 that are occurring during the printing process in comparison to just thermocycling the material. Although, they state that this part of ongoing or upcoming work, the reviewer recommend the authors to hypothesize speculation trying to explain the underlying phenomenon based on the available information.

We have added additional reasoning to the analysis and discussion to attempt to provide better explanation of our results.

  • There are 1 or 2 minor spelling errors throughout the paper and the thermogramm (Figure 4) is standing out since it seems to be plot from the measurement device itself rather than a postprocessed figure. If possible, please match to the style of the other diagrams.

In addition, they could improve on the formatting a bit, since the reader has to scroll through the document quite a lot to find the diagrams that are described in the text. This is something to be considered from a formatting point of view.

Figure 4 has been amended appropriately and typos have been corrected.

Reviewer 2 Report

The article presents an interesting analysis of the cooling process. Until now, this process has been omitted in scientific works. For a big plus deserves choosing a material with excellent properties and large use in industry. A wide range of temperature analysis was selected. My only remark is the following - I propose to more describe in introduction the known research results on the subject and the FDM and FFF technologies.

Author Response

We would like to sincerely thank all reviewers for taking the time to read our manuscript and provide constructive feedback. Based on these comments we have made a number of adjustments to improve the manuscript. All of these changes have been marked in yellow in the manuscript.

Reviewer 2

  • The article presents an interesting analysis of the cooling process. Until now, this process has been omitted in scientific works. For a big plus deserves choosing a material with excellent properties and large use in industry. A wide range of temperature analysis was selected.

Thank you.

  • My only remark is the following - I propose to more describe in introduction the known research results on the subject and the FDM and FFF technologies.

Further detail regarding the FFF technology, recent results in this field and the relation of this study to it has been added to the introduction.

Reviewer 3 Report

The article (Polymers-789505) focuses on the topic of crystallization of Nylon-12 caused by the fused filament fabrication. This topic is suitable for the journal of POLYMERS. In this article, the effect of different cooling rate and thermal gradient on crystallization of PA-12 were investigated via the DSC and WAXS measurements. The efforts are good. However, I have many concerns and suggest the authors revise their manuscript before its publishing.

  1. The title is “On the structural evolution of Nylon-12 samples due to the variation of 3D printing conditions”, however, in my impression, only crystallization was investigated in this article. The structural evolution is equal to the crystallization? The title is somewhat vague.
  2. The main contents of this article should include two aspects, the discussion of crystallization, and the discussion of FFF process. However, in this article, these two aspects are almost no relationship. If this article only investigates the non-isothermal crystallization of PA-12, it is difficult to be acceptable. The FFF process should be introduced in detail.
  3. It is possibly that the reader will feel difficult to find the function of this article, since there are no any results about the performance of FFF parts. Even, there is no any optimization and comparison of processing conditions. The key point is, which kind of structure is facilitated to the improvement of performance? What is the best processing condition? Both are not disclosed. Even if this article does not want to study the relative works of the performance/processing optimization by itself, a lot of discussions and introduction are still necessary. Even if the so-called structural evolution, it seems to be still unclear after reading this article.
  4. The thermal variation of the non-isothermal process as shown in Figure 1 may be greatly different from the really FFF process. The feasibility of using the non-isothermal crystallization to simulate the cooling conditions during FFF is really needed to be evaluated. Furthermore, the really FFF process is a dynamical process. It is well known that the flow-induced effect of FFF process may have an influence on the structural evolution. The ignoration of the dynamical condition may also influence the result. It is suggested to be considered more. At least, a lot of introduction and discussion should be added.
  5. The effect of FFF process condition should be well introduced in the background and/or the experimental section. Take the printing path as an example, which deciding the deposition orientation, is an important influencing factor in extrusion-based 3D printing, and I believe the author understand it. Many article have investigated the effect of fabrication processing conditions, such as Rapid Prototyping Journal, 2014, 20 (3): 192–204, and International Journal of Machine Tools and Manufacture, 2004, 44, 585–594, especially, the deposition-inducing effect exists for some materials, for example, Rapid Prototyping Journal, 2017, 23, 869-880, Industrial & Engineering Chemistry Research, 2019, http://dx.doi.org/10.1021/acs.iecr.9b04285, and Polymer Composites, 2020, 41, 60-72. Considering the references listed are obviously lacked, I therefore suggest the author mention this concern in the section of Introduction or experiment, although it may no influence for PA-12. At least, a lot of comments should be added into for making the paper more reasonable and objective, and the relevant references should be listed and commented. Furthermore, a detailed description of printing path is given, even if the path was automatically given by the controlling software of the printer. If the detailed processing condition is not clear, the effects are actually hardly to be fixed.
  6. Some minor revisions are suggested: 1’) The reference 1 is better to be a review article, considering the first sentence is mainly introduced the FFF process. 2’) More reference should be listed, there is now only 11 reference listed. 3’) the sub-title should be revised, since two section 4 appeared.  

Overall, this reviewer suggests the author major revises this article before its publication.

Author Response

We would like to sincerely thank all reviewers for taking the time to read our manuscript and provide constructive feedback. Based on these comments we have made a number of adjustments to improve the manuscript. All of these changes have been marked in yellow in the manuscript.

Reviewer 3

  • The article (Polymers-789505) focuses on the topic of crystallization of Nylon-12 caused by the fused filament fabrication. This topic is suitable for the journal of POLYMERS. In this article, the effect of different cooling rate and thermal gradient on crystallization of PA-12 were investigated via the DSC and WAXS measurements. The efforts are good.

Thank you.

  • The title is “On the structural evolution of Nylon-12 samples due to the variation of 3D printing conditions”, however, in my impression, only crystallization was investigated in this article. The structural evolution is equal to the crystallization? The title is somewhat vague.

Title changed to “Synchrotron X-ray scattering analysis of Nylon-12 crystallisation due to the variation of 3D printing conditions”

  • The main contents of this article should include two aspects, the discussion of crystallization, and the discussion of FFF process. However, in this article, these two aspects are almost no relationship. If this article only investigates the non-isothermal crystallization of PA-12, it is difficult to be acceptable. The FFF process should be introduced in detail.

A more detailed introduction to the FFF process has been added to the paper. The link between FFF processing conditions and non-isothermal crystallisation is enhanced. In addition, a refinement of the discussion of the scope of this study has been added.

  • It is possibly that the reader will feel difficult to find the function of this article, since there are no any results about the performance of FFF parts. Even, there is no any optimization and comparison of processing conditions. The key point is, which kind of structure is facilitated to the improvement of performance? What is the best processing condition? Both are not disclosed. Even if this article does not want to study the relative works of the performance/processing optimization by itself, a lot of discussions and introduction are still necessary. Even if the so-called structural evolution, it seems to be still unclear after reading this article.

Further details regarding the processing conditions have been added to the Methods section. A brief discussion regarding the processing conditions has been added to the Result Analysis section. However, the scope of this study does not extend to mechanical performance, which is to be addressed in a proceeding study. This study has its particular focus on the structural evolution as a function of cooling rate, and we believe that expanding its scope to address performance would represent an over-expansion of scope.

  • The thermal variation of the non-isothermal process as shown in Figure 1 may be greatly different from the really FFF process. The feasibility of using the non-isothermal crystallization to simulate the cooling conditions during FFF is really needed to be evaluated. Furthermore, the really FFF process is a dynamical process. It is well known that the flow-induced effect of FFF process may have an influence on the structural evolution. The ignoration of the dynamical condition may also influence the result. It is suggested to be considered more. At least, a lot of introduction and discussion should be added.

The dynamic nature of the FFF process certainly has an impact which cannot be ignored regarding the thermal gradients throughout the FFF part. Further introduction and discussion, as well as supporting references to existing research regarding the validity of the non-isothermal model for FFF has been added, which we hope clarifies this issue.

Furthermore, in Polymer 10 (2018) (doi:10.3390/polym10020168), Zhao et al discuss isothermal crystallisation as a basis for selective laser sintering of Nylon-12, which is a similarly complex and dynamical process. To this extent we believe there is a precedent for this choice of basis for study.

  • The effect of FFF process condition should be well introduced in the background and/or the experimental section. Take the printing path as an example, which deciding the deposition orientation, is an important influencing factor in extrusion-based 3D printing, and I believe the author understand it. Many article have investigated the effect of fabrication processing conditions, such as Rapid Prototyping Journal, 2014, 20 (3): 192Ð204, and International Journal of Machine Tools and Manufacture, 2004, 44, 585Ð594, especially, the deposition-inducing effect exists for some materials, for example, Rapid Prototyping Journal, 2017, 23, 869-880, Industrial & Engineering Chemistry Research, 2019, http://dx.doi.org/10.1021/acs.iecr.9b04285, and Polymer Composites, 2020, 41, 60-72. Considering the references listed are obviously lacked, I therefore suggest the author mention this concern in the section of Introduction or experiment, although it may no influence for PA-12. At least, a lot of comments should be added into for making the paper more reasonable and objective, and the relevant references should be listed and commented. Furthermore, a detailed description of printing path is given, even if the path was automatically given by the controlling software of the printer. If the detailed processing condition is not clear, the effects are actually hardly to be fixed.

Thank you for providing detailed links to these references. We have added these results to the introduction as means of better presenting the nature of the study. As above, we have also added details regarding path and processing conditions, in order to assuage concerns about the deposition orientation effect.

  • Some minor revisions are suggested: 1) The reference 1 is better to be a review article, considering the first sentence is mainly introduced the FFF process. 2) More reference should be listed, there is now only 11 reference listed. 3) the sub-title should be revised, since two section 4 appeared.

We have addressed these issues and made appropriate changes.

Round 2

Reviewer 3 Report

I recommend to accept this article in its present form.